# EV-miRNA-Mediated Intercellular Communication in the Breast Tumor Microenvironment

**DOI:** 10.3390/ijms241713085

**Published:** 2023-08-23

**Authors:** Francisca Sepúlveda, Cristina Mayorga-Lobos, Kevin Guzmán, Eduardo Durán-Jara, Lorena Lobos-González

**Affiliations:** 1Centro de Medicina Regenerativa, Facultad de Medicina, Clínica Alemana Universidad del Desarrollo, Santiago 7610615, Chile; fsepulvedag@udd.cl (F.S.); cristina.mayorga@ug.uchile.cl (C.M.-L.); kevin.guzman@ug.uchile.cl (K.G.); 2Advanced Center for Chronic Diseases (ACCDiS), Santiago 8380492, Chile; 3Facultad de Ciencias Químicas y Farmacéuticas, Universidad de Chile, Santiago 8380492, Chile; 4Subdepartamento de Genética Molecular, Instituto de Salud Pública de Chile, Santiago 7780050, Chile; eduran@ispch.cl

**Keywords:** extracellular vesicles, small EVs, cancer progression, miRNAs, tumor progression, metastasis

## Abstract

Cancer research has prioritized the study of the tumor microenvironment (TME) as a crucial area of investigation. Understanding the communication between tumor cells and the various cell types within the TME has become a focal point. Bidirectional communication processes between these cells support cellular transformation, as well as the survival, invasion, and metastatic dissemination of tumor cells. Extracellular vesicles are lipid bilayer structures secreted by cells that emerge as important mediators of this cell-to-cell communication. EVs transfer their molecular cargo, including proteins and nucleic acids, and particularly microRNAs, which play critical roles in intercellular communication. Tumor-derived EVs, for example, can promote angiogenesis and enhance endothelial permeability by delivering specific miRNAs. Moreover, adipocytes, a significant component of the breast stroma, exhibit high EV secretory activity, which can then modulate metabolic processes, promoting the growth, proliferation, and migration of tumor cells. Comprehensive studies investigating the involvement of EVs and their miRNA cargo in the TME, as well as their underlying mechanisms driving tumoral capacities, are necessary for a deeper understanding of these complex interactions. Such knowledge holds promise for the development of novel diagnostic and therapeutic strategies in cancer treatment.

## 1. Introduction

Recent advances in cancer biology have highlighted the significance of understanding the crosstalk between tumor cells and their neighboring microenvironment to unravel the mechanisms governing tumor growth and metastasis [1]. The tumor microenvironment (TME) comprises all the non-cancerous host cells in the tumor, including fibroblasts, endothelial cells, adipocytes, and adaptive and innate immune cells, as well as its non-cellular components, which include the extracellular matrix (ECM) and soluble products such as chemokines, cytokines, growth factors, and extracellular vesicles (EVs) [2].

During tumor growth, tumor cells dynamically interact with TME components, contributing to cancer cell survival, local invasion, and metastatic dissemination. For example, endothelial cells promote angiogenesis to counteract the hypoxic and acidic TME, ensuring an adequate oxygen and nutrient supply while removing metabolic waste [2]. Additionally, cancer-associated adipocytes exchange cytokines and lipids with tumor cells, leading to metabolic rewiring and the acquisition of proinflammatory and invasive phenotypes [3]. These interactions (among others), play essential roles in orchestrating the hallmarks of cancer [4], and underscore the importance of intercellular communication within the TME in their maintenance and promotion, which in the long term can promote tumor progression and metastasis [1].

Extracellular vesicles (EVs) have emerged as key mediators of intercellular communication. EVs are lipid bilayer-enclosed membranous structures secreted by all cells. They can be broadly classified into three subtypes: exosomes, microvesicles, and apoptotic bodies, depending on their biogenesis [5], and large (lEVs) and small (sEVs) EVs based on their size [6]. EVs carry diverse biomolecules, including proteins, lipids, and nucleic acids [7], which are necessary for cell-to-cell communication via cell–surface interactions [8]. This intercellular trafficking can occur in the form of paracrine signaling and/or with distant cells as a form of endocrine signaling [9]. Therefore, EVs can elicit pleiotropic responses in recipient cells during both physiological and pathological processes [10].

Of particular interest among the cargo components of EVs are microRNAs (miRNAs), a class of small non-coding RNAs that function in post-transcriptional regulation of gene expression. They are powerful regulators of various cellular activities including cell growth, differentiation, development, and apoptosis [11]. This highlights the importance of studying their role in cell-to-cell communication, particularly in pro-tumorigenic processes associated with the TME. Dysregulation of the cellular miRNA profile favoring the expression of tumor-promoter miRNAs (oncomiRs) and downregulating the expression of tumor-suppressor miRNAs have been widely associated with the acquisition of tumorigenic and pro-metastatic capacities [12,13]. Several studies have associated cellular miRNA dysregulation with enhanced tumor angiogenesis and vascular permeability [14,15], epithelial-mesenchymal transition (EMT) [16,17,18,19], and recently, adipose tissue remodeling [20,21,22]; processes that favor tumor growth, progression, and metastasis. However, the transport and delivery of miRNAs contained in EVs (EV-miRs) secreted by cells in the TME and their specific roles in the promotion of tumorigenic and pro-metastatic capacities have not been completely elucidated, and it is certainly an exciting and important research field.

In this review, we aimed to clarify the crucial role of EV-miRs in promoting pro-metastatic capacities, such as angiogenesis, vascular permeability and EMT in breast cancer (BC). In addition, we want to provide a better understanding of intercellular communication within the tumor microenvironment. We specifically delved into EV-miRs secreted by BC tumor cells and different cellular components of the BC TME, such as endothelial cells, cancer-associated fibroblasts (CAFs), and adipocytes. In this regard, we conclude this review by hypothesizing a role for cancer-associated adipocyte-secreted EVs and EV-miRs in favoring/promoting these processes and their relevance in BC, which could have important implications for the diagnosis, treatment, and development of targeted therapeutic approaches in BC.

## 2. BC Cell-Secreted EVs Promote Angiogenesis through the Delivery of Specific miRNAs

The intricate interplay between EVs-miRs and angiogenesis has been extensively studied in cancer. Angiogenesis, the process of new blood vessel formation, is a crucial phenomenon in cancer and typically occurs in hypoxic environments. It is primarily driven by soluble factors, including vascular endothelial growth factor (VEGF) and hypoxia-inducible factor 1-alpha (HIF-1α) [23]. In cancer, tumor angiogenesis is important for tumor nutrition and dissemination of tumor cells through the body. Evidence suggested an increase in EVs release in cancer cells in hypoxic environments [24]. Garcia-Hernandez et al., demonstrated that EVs secreted by the metastatic triple-negative MDA-MB-231 BC cells (TNBC) can mediate cellular processes involved in angiogenesis in HUVECs cells, which are human endothelial cells [25]. Furthermore, EVs from BC patients can also induce cellular processes involved in angiogenesis in HUVECs recipient cells [25,26]. The intricate interplay between EVs-miRs and angiogenesis has been extensively studied in cancer. Angiogenesis, the process of new blood vessel formation, is a crucial phenomenon in cancer and typically occurs in hypoxic environments. It is primarily driven by soluble factors, including vascular endothelial growth factor (VEGF) and hypoxia-inducible factor 1-alpha (HIF-1α) [23]. In cancer, tumor angiogenesis is important for tumor nutrition and dissemination of tumor cells through the body. Evidence suggested an increase in EVs release in cancer cells in hypoxic environments [24]. Garcia-Hernandez et al., demonstrated that EVs secreted by the metastatic triple-negative MDA-MB-231 BC cells (TNBC) can mediate cellular processes involved in angiogenesis in HUVECs cells, which are human endothelial cells [25]. Furthermore, EVs from BC patients can also induce cellular processes involved in angiogenesis in HUVECs recipient cells [25,26].

Among the large number of miRNAs involved in angiogenesis, only a few have been identified in cancer-derived EVs. However, the angiogenic effects of EVs contents have not yet been proven. In prostate cancer, EVs from five prostate cancer primary cell cultures were purified, miRNA patterns were analyzed by NGS, and miR-100-5p and miR-21-5p were the most abundant identified miRNAs [27]. Additionally, in colorectal cancer (CRC), hypoxic conditions promote increased expression of miR-155 in EVs cargo and inhibit FOXO3 in recipient cells [28]. Another important evidence was found in osteosarcoma, where a miRNA analysis of sEVs derived from different malignant human osteosarcomas revealed that miR-146a-5p was involved in the inhibition of osteoclast genesis and correlated with higher malignancy [29].

In the context of BC, there are a considerable number of articles related to the expression of pro-angiogenic EV-miRs. For example, miR-9 and miR-155 were among the overexpressed miRNAs in highly metastatic TNBC exosomes (sEVs) by RT-PCR, and a luciferase assay confirmed that the miRNAs target PTEN and DUSP14 expression [30,31,32]. Additionally, miR-27a expression in BC stem cells promotes its differentiation into cells with endothelial function and morphology, increasing angiogenesis in the TME. This EV-miR was also detected in high levels in BC EVs [33]. In addition, BC-derived exosomes (sEVs) promoted the activation of cancer-associated fibroblasts (CAFs) through the miR-146a/TXNIP axis to activate the Wnt pathway [34]. Moreover, after the purification of EVs derived from MDA-MB-231 and BT-474 BC cells, and the analysis of expression levels of EV-miRs, it was shown an increase in the miR-382, miR-21, and miR-210 content [24,35]. Coupled with that, Konoshenko et al., compared the expression levels of EV-miRs derived from BC patients and healthy females and reported an increased expression of miR-92a and miR-25-3p in BC patients [26]. Furthermore, Gonzalez-Villasana et al., determined exosomal small RNA expression levels by RT-PCR. Their results show that exosomes purified from the serum of BC patients and healthy donors contains miR-145, miR-155, and miR-382. Moreover, the results show the association between EVs concentration and the presence of breast cancer, creating the possibility to study how miRNAs packaged into exosomes play a role in BC progression [36].

Interestingly, several studies have reported a relationship between angiogenic processes and EV-miRs in different cancer types (Table 1). For instance, in nasopharyngeal carcinoma, increased exosomal miR-9 cargo inhibits angiogenesis by targeting MDK and through the PDK/AKT pathway. This data identifies the EVs-associated with the increased cargo of the miR-9 in tumorigenesis [37]. Moreover, miR-23a is also enriched in exosomes derived from nasopharyngeal cancer cells, and Bao et al., demonstrated that exosomal miR-23a modulated tube formation of HUVECs in vitro and affected blood vessel outgrowth in the zebrafish model by targeting TSGA10 [38]. Additionally, exosomes derived from hypoxic-hepatocellular carcinoma cells have increased levels of miR-23a and induce angiogenesis, whereas exosomes secreted by cells that overexpress miR-23a also induce angiogenesis in in vitro models by targeting SIRT1 in endothelial recipient cells [39]. In addition, CRC cells-derived EVs deliver and increased miR-21-5p content in endothelial cells. MiR-21-5p suppressed KRIT1 in recipient HUVECs and increases VEGFa, which promotes angiogenesis and vascular permeability in CRC [40]. Raimondi et al., also demonstrated that exosomal miR-21-5p and miR-148 potentiated tube formation of endothelial cells and increased angiogenic marker expression in osteosarcoma cells [41]. On the other hand, in lung cancer (LC) cell lines, the EVs promote HUVEC proliferation and migration in vitro by executing an important role in EVs-miR-23a targeting PTEN in recipient cells [42]. Kim et al., demonstrated enhanced tube formation in endothelial cells after stimulation with LC EVs-miR-494-3p by the Ras/syntenin-1 axis [43]. In addition, Zhang et al. suggested that exosomal-miR-221-3p derived from cervical cancer increases angiogenesis and proliferation of microvascular endothelial recipient cells by regulating MAPK10 [44]. Moreover, it is worth noting that tumor-associated macrophages are proven to be beneficial for angiogenesis. In this regard, Yang et al., confirmed the induction of angiogenesis in MAECs in vitro and increased vascular density in mice by M2 macrophage-derived exosomes and transfer of miR-155-5p and miR-221-5p levels in recipient cells [45].

As mentioned before, the literature has suggested the induction of angiogenesis through delivery of EV-miRs in different cancer types; however, it is important to note that there is a large amount of data focusing on BC. To illustrate, miR-182-5p is highly expressed in BC tissues, in the BC cell line MDA-MB-231 and in EVs secreted by MDA-MB-231 cells. Overexpression of miR-182-5p enhanced angiogenesis in vivo and in vitro by downregulating CMTM7 and activating the EGFR/AKT signaling pathway [46]. Additionally, miRNA-145 is highly present in EVs; however, in BC, STIM1 promotes angiogenesis by reducing exosomal miR-145 levels. Pan et al., used exosomes from STIM1-Knockout BC cells to probe the anti-angiogenic effect of exosomal miR-145 [47]. Furthermore, it is worth mentioning that Li et al., have shown that miR-210, a miRNA strongly related to angiogenesis, is increased in BC EVs cargo, and Gangadarn et al., have described its presence in mouse macrophages EVs together with miR-126 and miR-130, all related to the promotion of angiogenesis. They also demonstrated that macrophages EVs can induce tube formation in endothelial cells [48,49]. Ghaffari et al., suggested the induction of angiogenesis by EVs in BC cells, and identified that pro-angiogenic miR-9, -17-5p, -19a, -126, -130a, -132, -296, and -378 are increased in EVs, and also reported an increase in VEFG-A in recipient cells [50]. Likewise, Zheng et al., reveled a negative correlation between MCU and miR-4488 in BC EVs, they used EVs derived from downregulated MCU MDA-MB-231 cells, and the results suggested the relevant effect of miR-4488 suppressing angiogenesis in a nude mouse model [51].

**Table 1 ijms-24-13085-t001:** EV-miRs regulating angiogenesis in BC and other types of cancer.

miRNA	Source of EVs	Description	Reference
miR-182-5p	BC cells (MDA-MB-231)	miR-182-5p was highly enriched in EVs from BC cells and enhanced the proliferation, migration, and angiogenesis of HUVECs in vitro and in vivo EV-miR-182-5p promotes tumorigenesis and metastasis of BC cells	[46]
miR-4488	BC cells (MDA-MB-231)	MCU-dependent negative sorting of miR-4488 into EVs enhances angiogenesis in the metastatic niche	[51]
miR-145	BC cell (MDA-MB-231)	BC exosomes with lower levels of Ca^2+^ contain more miR-145, which targets IRS1 to exhibit an anti-angiogenic effect	[47]
miR-210 miR-126 miR-130	Mouse macrophages (Raw 264.7) BC cells (MCF-7)	Macrophage EVs can increase endothelial cell proliferation, migration, and tube formation in vitro	[49]
miR-210	BC cell lines	Hypoxic BC exosomes contained higher levels of miR-210	[24]
miR-21 miR-382	BC cells (MDA-MB-231, BT474)	Exosomes secreted from MDA-MB-231 and BT474 under DHA decreased expression levels of miR-21 and miR-382, and may inhibit angiogenesis	[35]
miR-92a miR-25-3p	Plasma from BC patients	Plasma-derived exosomes and total blood exosomes of BC patients had different expression levels of tumor-associated miR-92a and miR-25-3p and induced angiogenesis	[26]
miR-21-5p	BC cell lines and BC patients	miRNA is enriched in BC EVs	[52]
miR-9 miR-17-5p miR-19a miR-126 miR-130a miR-132 miR-296 miR-378	BC	Increase in pro-angiogenic EV-miRs results in tube formation in endothelial recipient cells	[50]
miR-21-5p	Colorectal cancer (CRC) cells	Activated β-catenin signaling pathway and increased VEGFa and Ccnd1	[40]
miR-221-3p	CRC cells	miR-221-3p from secreted EVs can promote endothelial cell proliferation, migration, and angiogenesis in vitro	[53]
miR-21-5p	Osteosarcoma cells	miRNA is overexpressed in the exosomal content and miRNA levels are increased in recipient cells. Promoted angiogenesis in vitro	[41]
miR-9	Nasopharyngeal carcinoma	Exosomal miR-9 secreted by nasopharyngeal carcinoma cells inhibits angiogenesis by targeting MDK and regulating the PDK/AKT pathway	[37]
miR-23a	Nasopharyngeal carcinoma	Exosomes transport miR-23a from nasopharyngeal carcinoma cells to endothelial cells, accelerating angiogenesis by directly targeting TSGA10 in vitro and in vivo	[38]
miR-23a	Hepatocarcinoma	miR23a from tumor cell colonies can induce angiogenesis by targeting SIRT1 in the recipient endothelial cells in vitro	[39]
miR-23a	Lung cancer (LC) cells	Lung cancer cell EVs transferred miR-23a into HUVECs, decreasing PTEN levels and increasing cellular proliferation and migration	[42]
miR-148 miR-21-5p	Osteosarcoma	Osteosarcoma cell exosomes increased expression of miR-21-5p and miR-148 in recipient cells. And both miRNAs promoted in vitro angiogenesis by stimulating the endothelial cells	[41]
miR-494-3p	LC cells	Ras/syntenin-1 axis may induce cancer progression by increasing miR-494-3p loading into sEVs in lung cancer cells	[43]
miR-155-5p	Tumor-associated macrophages (TAMs)	M2 macrophage-derived exosomes are enriched in miR-155-5p and miR-221-5p which promoted the angiogenic ability of endothelial cells	[45]
miR-21-5p	Endothelial progenitor cells	miR-21-5p was highly enriched in endothelial progenitor cell exosomes and suppressed the expression of an angiogenesis inhibitor THBS1 in the recipient endothelial cells	[54]

The miRNAs present in EVs derived from cancer cells can potentially participate in the communication between cells and TME, and likely modulate different pro-tumorigenic processes, including angiogenesis.

## 3. Tumor Cell-Secreted EVs Promote Endothelial Permeability through the Delivery of Specific miRNAs

Endothelial dysfunction, disruption and increased vascular permeability are essential processes favoring tumor nourishment, intra- and extravasation of tumor cells and metastasis. Several reports have demonstrated the role of EVs (mainly “exosomes” or sEVs) promoting those changes [55,56]. Some of them are focused on tumor cell-secreted EVs, however, there are also a few studies evaluating the role of cancer-associated fibroblasts (CAFs) [57], and immune cells-secreted EVs [58,59].

As mentioned, sEVs mediates phenotypic and functional changes in recipient cells through the delivery of their molecular cargo. One of the main components of EVs’ cargo are miRNAs (EV-miRs). EV-miRs can be actively delivered to endothelial cells through EVs to promote endothelial disruption and increase vascular permeability. Although several studies demonstrate an effect/role of tumor-derived EVs promoting/enhancing vascular permeability, not so many are focused on their miRNA cargo, and even less are focused on the causality mediated by specific EV-miRs in BC. In this regard, one of the first studies showing the importance of EV-miRs in the promotion of vascular permeability was done by Zhou W. and co-workers [15]. They showed that miR-105 is highly expressed in metastatic human BC cell lines and secreted in EVs. The transfer of EV-miR-105 from tumor cells to endothelial cells (HUVEC) promotes endothelial cell migration and destroys cell-cell junction through the downregulation of ZO-1 protein, which is essential in the maintenance of tight junctions. Interestingly, they showed that exosomes secreted by normal mammary epithelial MCF10A cells overexpressing miR-105 are also sufficient to increase endothelial cell migration and downregulation of ZO-1, discarding the possible effects of other component in the exosomes’ cargo. Their results were further confirmed in a murine model, where they showed that injection of miR-105-enriched EVs in immunosuppressed mice increase vascular permeability and promotes brain and lung metastasis. In the same line, another study showed that the transfer of “exosomal” miR-939 from triple-negative MDA-MB-231 BC cells to HUVEC endothelial cells caused the disruption of the endothelial barrier, mediated by the specific downregulation of VE-cadherin expression and the subsequent increase in vascular permeability and transendothelial migration of tumor cells [60]. Additionally, in an interesting and elegant study from 2015, Tominaga et al., showed that exosomal miR-181c secreted by BC cells can disrupt the brain blood barrier in vivo through the downregulation of PDPK1, promoting cofilin-induced actin remodeling and brain metastasis [61].

Besides BC, there are a few studies related to the delivery of EV-miRs in other cancer models. For instance, it was shown that LC cells subjected to hypoxia secrete significantly more EVs (exosomes) than cells in normoxia, and that those EVs contain higher levels of miR-23a. The authors showed that this enrichment in miR-23a was modulated by HIF-1α in CL1-5 LC cells. They also demonstrated that hypoxic EVs increase vascular permeability and trans-endothelial migration in vitro through the action of miR-23a over prolyl hydroxylase 1 and 2 (PHD1 and 2) and ZO-1 expression. Finally, they showed that inhibition of miR-23a action decreased tumor growth and angiogenesis in vivo, but vascular permeability was not assayed. Moreover, LC patients’ EVs contained elevated levels of miR-23a, and those EVs also increased in vitro vascular permeability [62]. Another recent study in LC cells showed that tumor-derived exosomal miR-3157-3p can be transported from tumor to endothelial cells to regulate permeability and angiogenesis. The authors showed that exosomes overexpressing miR-3157-3p can be incorporated by HUVECs and caused downregulation of TIMP/KLF2 which regulates VEGF/MMP2/MMP9 and occludin expression [63].

Finally, there are two interesting studies focused on CRC in which exosomal or EVs-associated miRs are involved in the promotion of vascular permeability and metastasis. In the first one, by regulating the expression of miR-25-3p, the authors showed that this miRNA can be delivered by exosomes to HUVEC cells to downregulate the expression of KLF2 and KLF4. This results in the inhibition of VEGFR, ZO-1, as well as Occludin and Claudin-5, which have important roles in the maintenance of tight and adherens junctions of endothelial cells. Endothelial disruption caused vascular permeability and angiogenesis in vitro, as well as CRC metastasis in the liver and lungs of the mice. Moreover, they showed that exosomes obtained from the blood of patients with metastasis have increased levels of miR-25-3p than patients without metastasis, which could be used as a possible biomarker of metastatic CRC [64]. Similarly, He and coworkers showed that exosomal delivery of miR-21-5p from CRC cells to endothelial cells increased vascular permeability through the direct downregulation of KRIT1 and the subsequent increase of b-catenin, VEGFa and Ccnd1 [40]. Figure 1 encompasses the main findings regarding EV-miRs modulating angiogenesis and vascular permeability in BC (Figure 1).

## 4. EV-miR Transfer Promotes EMT Acquisition in BC

The epithelial-mesenchymal transition (EMT) is a complex, gradual and not binary, cellular and molecular process, essential for tissue remodeling and metastases development [65,66,67,68,69]. Cells undergoing EMT upregulates the expression of molecules and transcription factors characteristics of a mesenchymal stage, which is reflected in the acquisition or increase in their invasive and migratory capabilities, cytoskeleton and extracellular matrix remodeling, promotion of their anchorage-independent growth capacity, among others changes [68,69,70,71]. The EMT is now considered an essential process in several steps of cancer progression, including tumor initiation, increase in tumor cell motility, invasiveness, intravasation, resistance to anoikis, and extravasation. Particularly in BC, EMT correlates with tumor aggressiveness, and it is associated with specific subtypes, with basal-like and claudin-low being the most EMT-associated subtypes [72,73,74]. As a complex cellular process, several molecules are implicated in the acquisition and maintenance of the EMT. Extracellular proteins such as TGF-β, Wnt ligands and growth factors can trigger an EMT. In addition, EMT-TFs expression and transcriptional regulation can further control the EMT program and phenotype. However, nowadays the attention is also focused on other interesting molecules, able to regulate EMT and other cellular processes such as miRNAs, particularly EV-miRs as important regulators of gene expression in recipient cells and as essential mediators of cell-to-cell communication in the BC TME.

There are a several studies regarding the EMT-promoting role of EV-miRs in cancer in general, however, this number is limited in the specific context of BC and are mainly focused on other cells in the BC TME rather than tumor cells-secreted EV-miRs itself. In one of the first studies, Singh et al., have reported an increase of the invasion capacity of mammary epithelial cells (as a hallmark of EMT); change mediated by the EV-miR-10b secreted by metastatic BC cells [75], via targeting and inhibition of HOXD10 and KLF4. Another study by Donnarumma et al., demonstrated a role of Exo-miRs produced by other cells in the TME such as fibroblasts [76]. This study described miR-21, -378, and -143 delivered by EVs to BC cells, promoting the EMT and increasing their mammosphere formation capacity and their anchorage-independent cell growth. Recently, it was also shown that BC-secreted exosomes can regulate invasion and metastasis via EV-miR146a. This study showed that EVs enriched in this miRNA can promote tumor cell invasion and EMT and can activate CAFs, via inhibition of its target TXNIP in fibroblasts [34]. Similarly, it was shown that BC cells-secreted EV-miR-370-3p can activate normal fibroblast into CAFs, which then promotes BC stemness, migration, invasion and EMT through the inhibition of CLYD and activation of the NF-kB pathway [77]. In an interesting study, Li et al., demonstrated that the delivery of miR-197 by BC stem-cells-secreted EVs promotes BC cells growth and EMT, by targeting and inhibiting PPARG expression [78]. Interestingly, Le et al., reported that the colonization ability of BC could be transferred via EVs from cells that colonize easily to those that colonize with difficulty [79]. The miR-200 family miRNAs, including miR-200a, miR-200b, miR-200c, miR-429 and miR-141, were secreted in EVs and transferred from cells that colonized easily to those that colonized with difficulty, which finally promoted mesenchymal-to-epithelial transition by targeting ZEB-2 and SEC23A in vitro. This study highlights the role of EV-miRs also promoting the inverse process, MET, which is essential for circulating and disseminating tumor cells to finally colonize and proliferate in distant tissues and generate metastatic foci. These and other studies implicating EV-miRs in the acquisition of EMT in BC and other types of cancer are listed in Table 2.

For instance, it has been shown that exosomes secreted by melanoma cells promotes the EMT on melanocytes through miR-191 and let-7a [83]. In an interesting study published in 2018, the Exo-miR profile of LC cells was identified, before and after inducing an EMT. It was found that Exo-miRs secreted by LC epithelial A549 cells (E-exosomes; E from Epithelial) differ from that of the same cells after treatment with TGF-β, thus after the induction of EMT (M-exosomes; M from Mesenchymal). Moreover, M-exosomes were able to induce EMT in recipient epithelial tumor and non-tumor cells, suggesting an important function of the Exo-miR profile transferred from mesenchymal-cell-derived exosomes to epithelial cells [84]. However, no functional assays were performed in order to elucidate specific causal roles of differentially EMT-enriched Exo-miRs (Figure 2).

## 5. Adipocytes as an Important Source of EVs in the TME

In cancer development, the tumor-cell intrinsic factors play an important role, however, interactions between the TME and tumor influence the biology and progression of the disease to a large extent as well [85]. Adipose tissue is composed of adipocytes, stromal vascular fraction, and pluripotent stem cells [86]. Adipocytes from different adipose tissues have metabolic and endocrine functions such as energy storage, release of free fatty acids, and secretion of adipokines, which are bioactive molecules including cytokines, hormones, and enzymes [87]. In addition to the classical adipokines, adipocytes produce and release EVs, which have emerged as crucial mediators of adipocyte intercellular communication [88].

Connolly et al., characterized EVs secreted pre- and post-adipogenesis in a 3T3-L1 cells model and found that production of EVs per cell are greater prior to adipogenic differentiation [89]. Also, they evidence differences in the composition of EVs secreted pre- and post-adipogenesis, but the functional effects of these EVs are poorly understood [89]. In a further investigation, the EVs secreted by adipocytes were quantitatively and qualitatively characterized through morphological and size parameters, sucrose flotation properties, protein patterns, and lipidomic analysis [90]. 3T3-L1 cells were differentiated for EVs production, and isolation by ultracentrifugation resulted in two distinct subpopulations: large extracellular vesicles (lEVs) and small extracellular vesicles (sEVs). Specific protein signatures were identified in these subpopulations of EVs, and lipidomic analysis revealed specific cholesterol enrichment in sEVs. In addition, they confirmed these results in primary murine adipocytes [90].

Following the characterization of EVs secreted by adipocytes (ADEVs), interest in the functional study of these vesicles has increased. Heretofore, research in ADEVs has demonstrated that they may play an endocrine and paracrine role in health and disease, due to the content of the EVs [91]. The packaging of proteins and miRNAs into ADEVs can be selective and reflect the microenvironment of the cell; for example, ADEVs released under disease conditions can exacerbate or drive pathologies associated with disease complications [91]. Recently, adipose tissue has been found as an important source of miRNAs in EVs [92]. This is evidenced by a substantial decrease in the levels of circulating exosomal miRNAs in an adipose-tissue-specific knockout of miRNA-processing enzyme Dicer mouse model, as well as in humans with lipodystrophy [92].

Microarray analysis of ADEVs secreted by differentiated 3T3-L1 cells revealed 143 miRNAs, among which miRNAs involved in adipose differentiation, such as let-7b, miR-103, miR-146b, and miR-148a, were significantly upregulated in differentiated cells [93]. One study on mesenchymal stromal/stem cells (MSCs) from porcine adipose tissue characterized the miRNA expression profile in EVs using RNA-seq and found four annotated miRNAs (miR148a, miR532-5p, miR378, and let-7f) predicted to modulate genes involved in apoptosis, angiogenesis, proteolysis, stem cell differentiation, cellular reprogramming, and cell growth [94].

A study of exosomes isolated from visceral and subcutaneous adipose samples collected from obese and lean patients revealed 88 mature miRNAs that were differentially expressed and 55 annotated miRNAs in obese visceral samples compared with lean visceral samples. They focus on miRNAs validated or predicted to target miRNAs in TGF-β and Wnt/β-catenin pathways (miR-23b, miR-148b, miR.182, miR-3681, miR-4269, miR-4319, and miR-4429) and finally they evaluated the uptake of exomes in lung epithelial cells. Nevertheless, to fully understand the function of miRNAs in ADEVs more studies will be needed [95].

The role of adipose tissue in tumor progression has been described and miRNAs contained in ADEVs can modulate fundamental biological capabilities for the development and progression of cancer, such as migration, metabolic reprogramming, angiogenesis, invasion, apoptosis, immune function, and proliferation [96]. For example, in the regulation of hepatocellular carcinoma (HCC), ADEVs can transfer miR-23a/b to hepatocellular cancer cells and promote cellular growth and migration [97]. The most recent study by Mathiesen et al., determine the contribution of omental (OM) and subcutaneous (SC) adipose tissue EVs (ATEVs) in metastatic prostate cancer (PC3ML cells) and evaluated the miRNA cargo with NanoString array. Forty-eight miRNAs were detected in SC ATEVs, 55 in OM ATEVs and 44 in both; transcriptomic analyses on PC3ML after treatment with ATEVs showed differentially expressed genes associated with proliferation, invasion and glycolysis and were validated with functional assays [98].

Studies on the interactions between adipose tissue and tumor cells have gained interest because some tumors start or metastasize near adipose tissue, and breast, prostate, and ovarian tumors are in sustained contact with adipose tissue [99]. Adipocytes surrounding cancer cells undergo phenotypic changes and are referred to as cancer-associated adipocytes (CAAs) [90,100]. CAAs secrete different soluble factors and extracellular vesicles that promote tumor progression [101,102,103]. Currently, there is limited evidence regarding EVs and EV-miRs secreted by CAAs. Au Yeung et al. performed the first study, using RNA-seq to identify higher levels of miR-21 in exosomes from CAAs derived from omental tissues of patients with high-grade serous ovarian cancer than in exosomes isolated from cancer-associated fibroblast tissue. Functional studies also revealed that miR-21 is transferred from exosomes to ovarian cancer cells and confers chemoresistance [101]. The miRNAs contained in EVs secreted by adipocytes are summarized in Table 3.

To date, most research has focused on EV-miRs secreted by adipocytes, and there is only one study on EV-miRs secreted by CAAs (Figure 3).

The primary cellular component in breast stroma is adipocytes, accounting for 56% of the non-lactating tissue [109]. In the BC microenvironment, due to the proximity between adipocytes and tumor cells, adipocytes and CAAs are involved in BC progression, and interest in understanding their mechanisms in BC has increased in recent years. To date, many studies have focuses on EVs secreted by adipocytes; however, further studies are required to characterize EVs secreted by CAAs and their miRNAs cargo to understand their potential role in mediating cell-cell communication in the TME in BC.

## 6. Conclusions

Cellular communication within the tumor microenvironment is mediated by extracellular vesicles and their cargo delivery. Notably, the miRNA content within these EVs contributes to cellular reprogramming and transformation, promoting various processes such as angiogenesis, endothelial permeability, epithelial–mesenchymal transition, and the conversion of adipocytes into cancer-associated adipocytes (Figure 4). Consequently, this communication within the TME plays a significant role in the acquisition and maintenance of cancer hallmarks. Studying EV-mediated processes in the context of breast cancer is crucial, as it holds the potential to uncover novel therapeutic alternatives, and miRNAs may serve as biomarkers and prognostic markers for patient monitoring. To date, there have been 399 clinical trials involving the use of miRNAs as biomarkers in cancer, 17 of which have focused on the study of miRNAs for the diagnosis and prognosis of breast cancer; however, more in-depth investigations will be necessary on the use of miRNAs contained in extracellular vesicles and their therapeutic applications. It is important to acknowledge that the precise functions of and mechanisms underlying this cellular communication are not yet fully understood, particularly in the case of cancer-associated adipocytes, which presumably have a key role in mediating the acquisition of tumor capacities. Gaining a comprehensive understanding of these processes will not only enhance our knowledge of cell communication in cancer but also lead to new therapeutic applications.

## Figures and Tables

**Figure 1 ijms-24-13085-f001:**
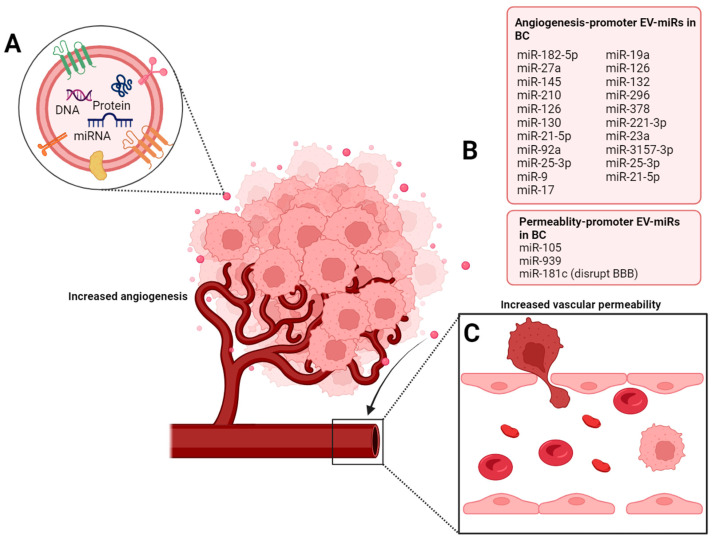
Delivery of EV-miRs secreted by tumor cells promotes vascular permeability and angiogenesis in BC. Tumor cells and other cells in the TME (**A**) secrete and deliver EV-miRs (**B**), the incorporation of which by endothelial recipient cells favors the disruption of endothelial integrity (**C**), vascular permeability, and angiogenesis, which can finally end in tumor nourishment, tumor cell intravasation, and dissemination. EV-miRs involved in the regulation of angiogenesis and permeability are listed in the figure and include EV-miR-182-5p, -210, -126, 92a, -9, -105, and 939, among others.

**Figure 2 ijms-24-13085-f002:**
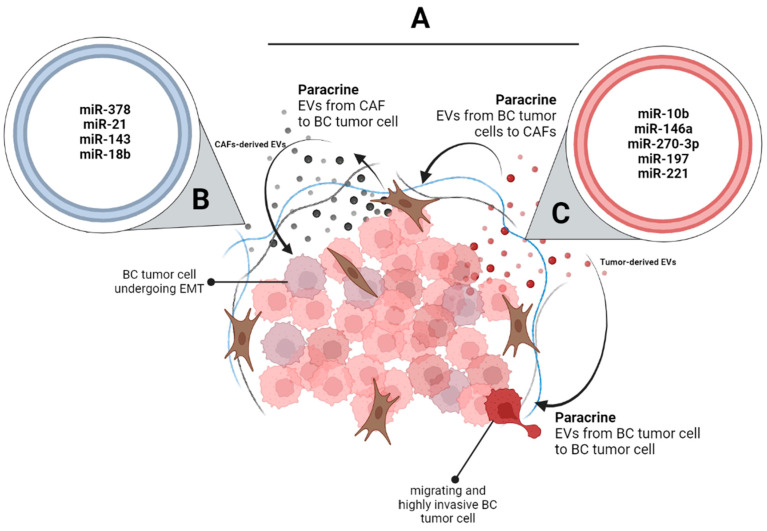
BC tumor-derived or/and CAF-derived EVs promote EMT in BC tumor recipient cells. (**A**) High heterogeneity in a primary BC tumor is important when considering cell–cell communication and crosstalk between different cell types and between tumor cells with different properties and transformations. BC-derived EV-miRs can promote EMT in less aggressive BC recipient cells in a paracrine manner. (**B**) Among the EV-miRs implicated in these changes are miR-378, miR-21, miR-143, and miR-18b. On the other hand, CAFs in the BC TME also secrete EVs and EV-miRs and have a role in the promotion of EMT. (**C**) EV-miR-10b, -miR-146a, and -miR-221, among others, are secreted by CAFs and can promote the acquisition of EMT in BC tumor cells.

**Figure 3 ijms-24-13085-f003:**
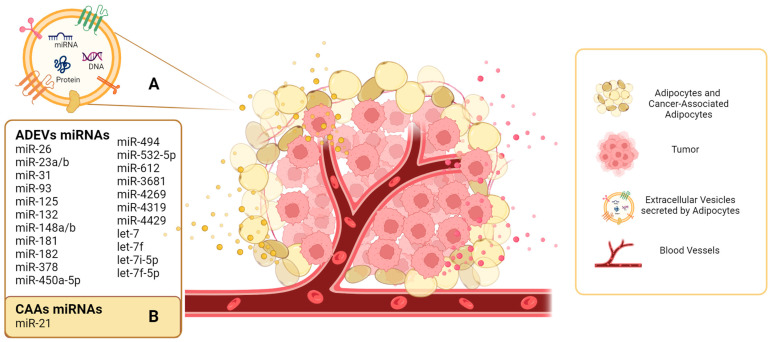
Extracellular vesicles secreted by adipocytes and miRNA cargo. (**A**) The extracellular vesicles secreted by adipocytes (ADEVs) are represented in yellow, and the miRNAs described to date are shown in the yellow rectangle. (**B**) In the context of the tumor microenvironment, these adipocytes are transformed into cancer-associated adipocytes (CAAs), and the presence of miR-21 in EVs secreted by CAAs has been described.

**Figure 4 ijms-24-13085-f004:**
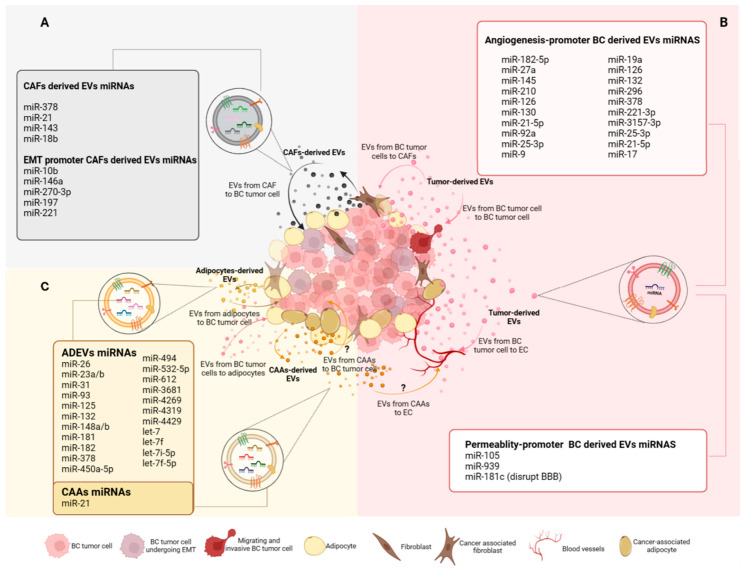
Heterogeneity in primary breast cancer tumor and EV-miRNA-mediated cell–cell communication. The tumor microenvironment is heterogeneous due to the presence of different types of cells and secreted factors. Among the secreted factors, extracellular vesicles and their miRNA cargo contribute to cell–cell communication and cancer progression. (**A**) CAF-derived EV-miRs can promote EMT in BC. (**B**) BC-derived EV-miRNA facilitate endothelial permeability and promote angiogenesis within the BC microenvironment. (**C**) ADEV miRNAs can interact with BC tumor cells and promote BC progression. Adipocytes are transformed into CAAs and only the presence of miR-21 in secreted EVs has been described. To date, the effect of CAA EV-miRNAs and whether these EVs impact ECs remains unknown. The signal “?” represents cellular communication mediated by extracellular vesicles and miRNAs, which is unknown and requires further investigation. ADEVs: adipose-derived extracellular vesicles; BBB: blood–brain barrier; BC: breast cancer; CAAs: cancer-associated adipocytes; CAFs: cancer-associated fibroblasts; EC: endothelial cells; EMT: epithelial–mesenchymal transition; EVs: extracellular vesicles.

**Table 2 ijms-24-13085-t002:** EV-miRs implicated in the promotion of EMT in BC and other types of cancer.

miRNA	Source of EVs	Description	Reference
miR-10b	Metastatic BC cells	miR-10b-enriched EVs secreted by metastatic BC cells promote invasion in mammary epithelial cells by inhibiting HOXD10 and KLF4	[75]
miR-21 miR-378 miR-143	Fibroblasts (CAFs)	EVs secreted by CAFs delivered to tumor cells promote EMT, mammosphere formation, and anchorage-independent growth	[76]
miR-146a	BC cells	EV-miR-146a regulates invasion and EMT, and activates fibroblasts through inhibition of TXNIP	[34]
miR-270-3p	BC cells	EV-miR-370-3p activates normal fibroblasts into CAFs, promoting BC stemness, migration, invasion, and EMT through the inhibition of CLYD	[77]
miR-197	BC stem cells	EV-miR-197 promotes BC cells growth and EMT, by targeting and inhibiting PPARG expression	[78]
miR-221	TNBC cells	Microvesicle delivery of miR-221 enhances proliferation, survival, and EMT in non-metastatic recipient tumor cells. It also enhances metastasis in vivo in a PTEN-Akt-Nf-kB-dependent fashion	[80]
miR-18b	CAFs	CAFs-derived exosomes enriched in miR-18b promote BC cell invasion and metastasis by targeting TCEAL7	[81]
miR-223	TAMs	Microvesicles secreted by IL-4-activated macrophages (M2 TAMs) deliver miR-223 to BC recipient cells to promote invasion via the Mef2c/β-catenin pathway	[82]
let-7i miR-191 let-7a	Melanoma cells	Exosomes from melanoma cells containing high levels of let-7i modulate an EMT-resembling process in primary melanocytes	[83]

**Table 3 ijms-24-13085-t003:** miRNAs in extracellular vesicles secreted by adipocytes.

miRNAs	Source of EVs	Description	References
miR-148a let-7f miR-532-5p miR-378	Porcine adipose-derived stem cells	Characterized the cargo of EVs by high-throughput RNA sequencing. At least 386 annotated miRNAs were read but four were enriched in EVs.	[94]
miR-23b miR-148b miR-182 miR-3681 miR-4269 miR-4319 miR-4429	Visceral and subcutaneous adipose samples from obese and lean patients	Compared miRNA levels between obese and lean visceral exosomes.	[95]
miR-31	Human ADSCs	miR-31 promote angiogenesis in HUVECs by targeting factor-inhibiting HIF-1 (FIH1).	[104]
miR-21	Normal and cancer-associated adipocytes from ovarian cancer patients	miR-21 is transferred from cancer-associated adipocytes to cancer cells and confers chemoresistance.	[101]
miR-450a-5p	Rat adipose tissue and ADSCs	miR-450a-5p mediates adipogenic differentiation.	[105]
miR-132	Human adipose-derived stem cells (ADSCs)	miR-132 was transferred from ADSCs to lymphatic endothelial cells and promoted proliferation, migration, and tube formation.	[106]
miR-23a/b	3T3-L1 cells, serum, and tumor tissues of hepatocellular carcinoma patients.	miR-23a/b was upregulated in serum exosomes and tumor tissue. Results suggested that miR-23a/b was derived from adipocytes and transported into cancer cells, conferring chemoresistance.	[97]
let-7	Human ADSCs	Human ADSC-EVs contribute to angiogenesis via let-7.	[107]
let-7i-5p, let-7f-5p	Human ADSCs	Human ADSC-EVs promote migration and invasion of endothelial cells.	[108]
miR-93, miR-125, miR-16, let7, miR-612, miR-494, miR-181	Human adipose tissue	miRNAs contained in ADEVs upregulate genes that may impact increased proliferation and deregulate genes that reduce invasion of prostate cancer cells.	[98]

## Data Availability

Not applicable.

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
