# Peer review of "EV-miRNA-Mediated Intercellular Communication in the Breast Tumor Microenvironment"

_ijms, 2023, doi:10.3390/ijms241713085_

Round 1

Reviewer 1 Report

The authors, Sepúlveda F. et al., aimed to examine the role of extracellular vesicle-derived microRNAs (EV-miRs) in facilitating pro-metastatic processes, including angiogenesis, vascular permeability, and epithelial-mesenchymal transition (EMT), within the breast tumour microenvironment. This objective was addressed in their review article titled "EV-miRNAs-mediated intercellular communication in the breast tumour microenvironment." This constitutes a comprehensive and interesting topic of review work that is appreciated. However, minor concerns can be resolved to improve the quality of the manuscript, listed below.

1. The introduction should provide a more comprehensive discussion of how your study contributes new significance to the existing body of research. The authors' thought process regarding adipocyte remodeling, EV secretion, and the miRNA profile appears convoluted, leading to confusion among readers. The authors should certainly consider revisiting the introduction section.

2. The discussion is well written.

3. There is space for grammatical improvement.

The user's text could benefit from grammatical improvements and sentence reconstruction in certain areas, as some portions are difficult to understand.

Author Response

Thank you very much for your kind and assertive advice and comments to improve our review “EV-miRNAs-mediated intercellular communication in the breast tumor microenvironment”. This is the detail of the corrections we have made in each comments.

Reviewer 1

  1. The introduction should provide a more comprehensive discussion of how your study contributes new significance to the existing body of research. The authors' thought process regarding adipocyte remodeling, EV secretion, and the miRNA profile appears convoluted, leading to confusion among readers. The authors should certainly consider revisiting the introduction section.

Answer: Thanks for the comments because now we expect a version more comprehensive for the readers. The introduction was restructured with new way to paraphrasing and include new ideas with the intention for clarify to the readers, in pag 2 and 3.

  1. There is space for grammatical improvement.

Answer: Thanks for the comments, the grammatical forms was checked and analyzed.

Reviewer 2 Report

The authors summarized the recent findings about the role of miRNA encapsulated in extracellular vesicles (EVs) in breast cancers. The manuscript is well organized and suitable for publication.

Please address the following points:

1. Line 105-116.

Same sentences are duplicated.

2. Line 117-147

A little bit redundant. It seems that a lot of studies which is not related to angiogenesis of breast cancer are cited.

3. It is worth discussing about not only the biological roles but clinical significance of EV-miRNAs. 

Author Response

Reviewer 2

  1. Line 105-116: Same sentences are duplicated.

Answer: We eliminated the mentioned sentences, pag 3.

  1. Line 117-147: A little bit redundant. It seems that a lot of studies which is not related to angiogenesis of breast cancer are cited.

Answer: We eliminated the mentioned studies which is not related to angiogenesis of breast cancer are cited. Pag 3

  1. It is worth discussing about not only the biological roles but clinical significance of EV-miRNAs. 

Answer: We include the information reference to clinical trials with the studies involved the miRNAs with the projection in cancer diagnostic and prognosis. Pag 12
